# Comparative Outcomes of Respiratory Failure Associated with Common Neuromuscular Emergencies: Myasthenia Gravis versus Guillain–Barré Syndrome

**DOI:** 10.3390/medicina55070375

**Published:** 2019-07-15

**Authors:** Anantha R. Vellipuram, Salvador Cruz-Flores, Mohammad Rauf A. Chaudhry, Prashanth Rawla, Alberto Maud, Gustavo J. Rodriguez, Darine Kassar, Paisith Piriyawat, Mohtashim A. Qureshi, Rakesh Khatri

**Affiliations:** 1Department of Neurology, Paul L. Foster School of Medicine, Texas Tech University Health Sciences Center El Paso, El Paso, TX 79905, USA; 2Department of Internal Medicine, Hospitalist Sovah Health, Martinsville, VA 24112, USA

**Keywords:** mechanically ventilated GBS, mechanical ventilated MG, in-hospital mortality, length of stay, disability at discharge

## Abstract

*Background and objectives:* Myasthenia gravis (MG) and Guillain–Barré Syndrome (GBS) are autoimmune neuromuscular disorders that may present as neuromuscular emergencies requiring mechanical ventilation and critical care. Comparative outcomes of these disease processes, once severe enough to require mechanical ventilation, are not known. In this study, we compared the patients requiring mechanical ventilation in terms of in-hospital complications, length of stay, disability, and mortality between these two disease entities at a national level. *Materials and Methods:* Mechanically ventilated patients with primary diagnosis of MG (n = 6684) and GBS (n = 5834) were identified through retrospective analysis of Nationwide Inpatient Sample (NIS) database for the years 2006 to 2014. *Results:* Even though mechanically ventilated MG patients were older (61.0 ± 19.1 versus 54.9 ± 20.1 years) and presented with more medical comorbidities, they had lower disease severity on admission, as well as lower in-hospital complications sepsis, pneumonia, and urinary tract infections as compared with GBS patients. In the multivariate analysis, after adjusting for confounders including treatment, GBS patients had significantly higher disability (odds ratio (OR) 15.6, 95% confidence interval (CI) 10.9–22.2) and a longer length of stay (OR 3.48, 95% CI 2.22–5.48). There was no significant difference in mortality between the groups (8.45% MG vs. 10.0% GBS, *p* = 0.16). *Conclusion:* Mechanically ventilated GBS patients have higher disease severity at admission along with more in-hospital complications, length of stay, and disability compared with MG patients. Potential explanations for these findings include delay in the diagnosis, poor response to immunotherapy particularly in patients with axonal GBS variant, or longer recovery time after nerve damage.

## 1. Introduction

The most common neuromuscular emergencies encountered in the critical care units are myasthenic crisis and Guillain–Barré syndrome. Myasthenia gravis (MG) and Guillain–Barré syndrome (GBS) are autoimmune disorders of peripheral nervous system, which affect either the neuromuscular junction or the roots and nerves. They have an estimated prevalence at 14 to 20 per 100,000 population [1] and 1.2 to 3 per 100,000 population, respectively, in the United States [2]. Respiratory and pharyngeal muscles are usually involved, leading to ventilatory failure with hypoxemia and CO_2_ retention, requiring endotracheal intubation and mechanical ventilation. 

Although data are limited, the proportion of patients with MG who experience at least one episode of respiratory failure is as high as 10% to 20% [3], and its annual risk is approximately 2% to 3% [4]. Similarly, respiratory insufficiency occurs in up to 25% of patients with GBS, and 60% of them develop major complications like sepsis, pneumonia, and gastrointestinal bleeding [5]. Despite optimum care, the extent of recovery following an episode of respiratory failure can vary a lot among patients with MG or GBS, and it may depend on different factors such as the age of the patients, the presence of other medical comorbidities, and the severity of the symptoms at admission [6]. Thus far, how these two disease entities may differ in terms of the outcome following an episode of respiratory failure requiring mechanical ventilation is not reported. Therefore, we compared mechanically ventilated patients with MG and GBS at the national level in terms of in-hospital complications, length of stay, disability at discharge, and in-hospital mortality.

## 2. Patients and Methods

### 2.1. Data Source and Cleansing

We conducted a retrospective analysis of 2006–2014 hospital discharge information from the Healthcare Cost and Utilization Project-Nationwide Inpatient Sample (HCUP-NIS) administrative database. The NIS database is a stratified sample of inpatient admissions from 1000 acute care hospitals maintained by Agency of Health Care Quality and Research [7]. International Classification of Diseases, 9th edition—Clinical Modification (ICD-9-CM) codes for primary diagnoses of GBS (357.0) and MG crisis (358.01) were used to identify our initial cohort. We identified the mechanically ventilated GBS and MG patients using the ICD-9-CM codes: 93.90 and 96.04 as secondary diagnosis. ICD-9 procedure codes were used to determine the use of plasma exchange (PLEX) or plasmapheresis (99.71, 99.76), and intravenous immunoglobulin (IVIG) (99.14). 

To avoid double representation of the same patient, we excluded those patients whose disposition or admission type indicated a transfer to or from another short-term hospital. Patients with a hospital charge of less than $100 were likely coded incorrectly and were also excluded from the analysis. Similarly, patients with a negative length of stay or length of stay exceeding 365 days were eliminated from the dataset [7]. This study was deemed exempt by the Texas Tech University Health Sciences Center El Paso’s Institutional Review Board (El Paso, TX, USA), as HCUP-NIS is a public database with no personal identifying information [7].

Ethical approval for research: No animals were involved in our study. This study was deemed exempt by the Texas Tech University Health Sciences Center El Paso’s Institutional Review Board (El Paso, TX, USA), as HCUP-NIS is a public database with no personal identifying information. 

### 2.2. Independent and Outcome Variables

Independent demographic variables included patient age; gender; ethnicity (grouped into white, black, Hispanic, others); median household income (grouped into 0 to 25th percentile, 26th to 50th percentile, 51st to 75th percentile, and 76th to 100th percentile); hospital characteristics including bed size, location, teaching status, and region; insurance status; medical co-morbidities, that is, hypertension, diabetes mellitus, congestive heart failure, coagulopathy, chronic lung disease, renal failure, and chronic alcohol abuse, as obtained from Agency for Health Research and Quality comorbidity data files. Medical co-morbidities, that is, dyslipidemia, atrial fibrillation (AF), and smoking, were determined using the ICD-9 CM secondary diagnosis codes (272.4), (427.31 and 427.32), and (305.10–305.19), respectively. 

ICD-9 secondary codes were used to identify in-hospital complications such as pneumonia (486, 481, 482.8, and 482.3), urinary tract infection (590.0, 590.9) sepsis (995.91, 995.92, 996.64, 038, and 999.3), and in-hospital procedures such as gastrostomy (43.11) and red blood transfusions (99.04).

All Patients Refined Diagnosis Related Groups (APRDRG) severity index was used to measure disease severity at admission, and disability was classified as none-to-minimal, moderate, and severe. We used the variable “DISPUNIFORM” obtained from Agency for Health Research and Quality comorbidity data files to define none to minimal disability (routine, home health care (HHC), and against medical advice (AMA) and moderate to severe disability (transfer other: includes skilled nursing facility (SNF), intermediate care facility (ICF), another type of facility) [8]. 

We determined the length of stay (LOS), hospital charges, and in-hospital mortality using the variables “LOS”, “TOTCHG”, and “died” from Agency for Health Research and Quality comorbidity data files [8]. Prolonged length of stay was defined as LOS ≥ 7 days.

### 2.3. Statistical Analysis

The SAS 9.4 software (SAS Institute, Cary, NC, USA) was used to convert NIS database data into weighted counts to generate national estimates, following Healthcare Cost and Utilization Project recommendations. We performed univariate analysis, chi-square for categorical, and *t* test for continuous variables to identify differences in study variables and outcome end points between mechanically ventilated GBS and mechanically ventilated MG patients.

For multivariate analysis, two logistic regression model were created. Model 1 was used to assess the difference in terms of disability at discharge, included alive patients only, and was adjusted for age (as a continuous variable), gender (as a categorical variable), medical comorbidities (as a categorical variables), in-hospital complications (as a categorical variables), APRDRG severity scale (as a categorical variable), insurance status (as a categorical variable), and median household income for patient’s ZIP code (as a categorical variable), which were significant (*p* ≤ 0.05) in univariate analysis. Model 2 was used to assess the difference in terms of prolonged length of stay (PLOS) at discharge; included all patients; and was adjusted for age (as a continuous variable), gender (as a categorical variable), medical comorbidities (as a categorical variables), in-hospital complications (as a categorical variables), APRDRG severity scale (as a categorical variable), insurance status (as a categorical variable), and median household income for patient’s ZIP code (as a categorical variable), which were significant (*p* ≤ 0.05) in univariate analysis.

A comparative analysis of the outcome between the two groups of patients was performed considering demographic (age and sex) and clinical characteristics of the patients (medical comorbidities, in-hospital complications, severity of disability at admission).

Multivariate analysis was performed adjusting for age, gender, and potential confounders, which, in our case, included significantly different medical co-morbidities, in-hospital complications, and IVIG PLEX. Odds ratio (OR) of 95% CI was used to analyze the differences between the two groups in the multivariate analysis. OR = 1, no differences; OR > 1, the condition is more likely to occur in the GBS group; OR < 1, the condition is less likely to occur in the GBS group. A *p* value < 0.05 was considered statistically significant.

## 3. Results

### 3.1. Demographic and Clinical Characteristics of Mechanically Ventilated Patients with Guillain–Barre Syndrome (GBS) and Myasthenia Gravis (MG)

Between the years 2006–2014, a total of 5834 patients with GBS and 6684 with MG required mechanical ventilation (Table 1). We observed that patients affected by MG were older compared with GBS patients (61.0 ± 19.1 years vs. 54.9 ± 20.1 years, respectively; *p* < 0.0001), and the incidence of respiratory failure was higher in women with MG than those with GBS (50.5% vs. 42.0%; *p* ≤ 0.0001). This observation is consistent with the fact that while GBS can strike at any age, although it is more frequent in adult and older people and both sexes are equally prone to the disorder [6], the age at MG onset is bimodal: there is a peak at the second and third decades with women, being more affected than men (3:1 ratio), while the second peak occurs during the sixth and seventh decades; is gender independent; and has a more severe clinical pattern than the early onset [9,10,11,12].

MG patients had significantly higher medical co-morbidities compared with GBS patients, which included diabetes mellitus, congestive heart failure, atrial fibrillation, coagulopathy, chronic lung disease, and dyslipidemia. GBS patients had greater alcohol abuse and smoking. Measurement of disease severity at the time of admission with APRDRG severity index demonstrated overall higher disease severity for GBS patients. In addition, 74.1% of mechanically ventilated GBS had severe disability compared with 66.6% of mechanically ventilated MG patients (*p* ≤ 0.0001). The total hospital charges were significantly higher for patients with GBS compared with MG ($298,378 ± 253,167 vs. $176,220 ± 163,584, *p* < 0.0001).

### 3.2. Comparative Analysis of In-Hospital Complications and Treatments 

We observed that the major complications of sepsis, urinary tract infection, and pneumonia developed more often in patients with mechanically ventilated GBS than with MG (Table 2). A significantly higher number of patients with GBS than with MG received gastrostomy (33.4% vs. 11.9%, respectively; *p* < 0.0001). 

Intravenous immunoglobulin (IVIG) and plasmapheresis (or plasma exchange, PLEX) are the two main immunotherapy treatments for both entities [13,14,15,16]. Both treatments are equally effective, and the combination has neither an additive nor a synergic action [17]. We observed that there was no difference in terms of rate of treatment with IVIG and/or PLEX between the two groups (58.1% vs. 55.8%, MG and GBS, respectively; *p* = 0.24). 

### 3.3. Comparative Analysis of the Outcomes 

We observed that mechanically ventilated GBS patients were hospitalized for a longer time compared with mechanically ventilated MG patients (25.6 ± 21.8 days vs. 15.7 ± 12.9 days, *p* = 0.001) and a higher number of GBS patients had a prolonged length of stay (≥ 7 days) (95.6% vs. 82.5%, *p* < 0.0001). Following remission from respiratory failure, a significantly higher number of GBS patients showed moderate to severe disability compared with MG (83.2% vs. 35.3%, *p* < 0.0001), whereas in-hospital mortality was the same between the two groups (10.0% GBS vs. 8.45% MG, *p* = 0.16). Finally, the multivariate analysis, after adjusting for age, gender, and potential confounders, showed that patients with GBS had significantly higher disability at discharge (OR 15.6; 95% CI 10.9–22.2) and longer length of stay (OR 2.48; 95% CI 2.22–5.48) (Table 3).

## 4. Discussion

The use of PLEX/IVIG and improved methods of mechanical ventilation in specialized critical care units has substantially decreased mortality rates in critically ill MG and GBS patients [18,19,20,21,22,23]. Despite current available medical treatment for symptom alleviation and immune modulation for MG [24], the risk of respiratory failure with the need for mechanical ventilation is still as high as 20% [3]. In GBS, despite immunotherapy, mechanical ventilation is needed in about 25% of patients [6].

Several studies have independently reported the outcomes of MG and GBS following respiratory failure [25,26,27,28,29,30]; however, a comparative analysis of the outcomes between these two disease entities in the United States has not been reported. In our study, we identified that mechanically ventilated GBS patients have higher in-hospital complications and length of stay, higher cost for the hospitalization, and higher disability at discharge compared with MG patients. The adjusted multivariate analysis showed that this difference cannot be imputable to the demographical and clinical features of the patients only, but to other factors too. Factors potentially leading to these differences include the possibility of diagnosis delays in GBS and/or simply the fact that with severe GBS and axonal injury, the recovery time is much longer than the reversal of myasthenic crisis. This is to say that the GBS variant may have an impact on the clinical course and outcome [31,32]. From these variants, which include acute inflammatory demyelinating polyradiculoneuropathy (AIDP), the most common form in the United States; Miller Fisher syndrome (MFS), which occurs in about 5% of people with GBS in the United States; and other axonal variants including acute motor axonal neuropathy (AMAN) and acute motor-sensory axonal neuropathy (AMSAN), the response may be poorer for those variants with axonal injury [33].

Timely initiation of ventilator support is critical for better outcomes in neuromuscular respiratory failure, and the optimal timing of intubating patients with respiratory failure is sometimes challenging as premature intubation or emergency intubation carry their own disadvantages [31,32]. Endotracheal intubation can carry more risks in GBS than in other patients requiring emergency airway control, because of dysautonomia, which can cause hypotension or cardiac dysrhythmias during airway manipulation [34]. As 60% of those who are intubated develop major complications, including pneumonia, sepsis, gastrointestinal bleeding, and pulmonary embolism are, in a way, not surprising to see a higher rate of complications among GBS, as they have the longer LOS and ventilation days [5,35,36]. It can be argued that PLEX/IVIG is the critical factor in limiting the LOS and ventilation days and, therefore, the risk of complications; however, it is also likely that the reversibility of, not only the immune process, but also the reversibility of the injury resulting from the immune process, are important determinants in the outcomes measured in this study.

### Limitations

Our study has several limitations related to the utilization of an extremely large retrospective database with a predefined study cohort [37]. NIS database does not capture the mortality events and administration of PLEX and IVIG taking place in outpatient settings before hospital admission, as these patients are likely to represent a different population with different disease severity and treatment indication [7,37]. The analysis for each patient was limited to single hospitalization, as a result, information about the long term outcomes such as rates of readmission and survival cannot be determined [7]. We used ICD-9 codes to identify mechanically ventilated GBS and MG patients, the validity of which has been questioned before [38], however, the coding errors follow a random distribution and do not manifest in systemic fashion, and thus it is unlikely to impact the statistical results because they are buffered by the size of NIS database [37]. Disease severity measurement by clinical assessment, serum concentration of acetylcholine receptor and muscle-specific kinase autoantibodies, and specific treatment regimens are not known. In addition, the information about administration of corticosteroids and other immunosuppressive agents cannot be determined using this database [7]. Despite these limitations, the large sample size and the fact that the dataset is a reflection of current clinical practice provide the best argument to say the findings are real and deserve further study.

## 5. Conclusions

On the basis of the analysis of the NIS data, despite no difference in mortality, mechanically ventilated GBS patients have higher in-hospital complications, length of stay, and moderate to severe disability compared with MG patients. This may reflect a delay of the diagnosis of GBS, poor response to immunotherapy in certain GBS variants, or the degree of reversibility of the injury caused by the disease process.

Generalizability: The large sample size of NIS database means that generalizing our study findings gives a very likely a reflection of current clinical management and outcomes of mechanically ventilated MG and GBS patients. 

## Figures and Tables

**Table 1 medicina-55-00375-t001:** Demographic and clinical features of mechanically ventilated patients with Guillain–Barré syndrome (GBS) and Myasthenia gravis (MG). APRDRG, All Patients Refined Diagnosis Related Groups.

	Mechanically Ventilated GBS	Mechanically Ventilated MG	*p*-Value
Number of patients	5834	6684	
Age (mean years ± SD)	54.9 (±20.1)	61.0 (±19.1)	<0.0001
Gender (%)			
Women	2452 (42.0%)	3375 (50.5%)	<0.0001
Ethnicity (%)			
White	3808 (74.5%)	3951 (65.6%)	<0.0001
Black	426 (8.33%)	1164 (19.3%)	
Hispanic	527 (10.3%)	596 (9.90%)	
Asian or Pacific Islander	116 (2.27%)	135 (2.24%)	
Native American	50 (0.98%)	15 (0.25%)	
Other	182 (3.57%)	161 (2.68%)	
Income by zip code (%)			
0–25th percentile	1417 (25.1%)	1888 (29.0%)	0.04
26th to 50th percentile (median)	1363 (24.1%)	1692 (26.0%)	
51st to 75th percentile	1422 (25.2%)	1475 (22.6%)	
76th to 100th percentile	1442 (25.5%)	1459 (22.4%)	
Insurance status			
Medicare	2157 (37.1%)	3799 (56.9%)	<0.0001
Medicaid	765 (13.1%)	849 (12.7%)	
Private insurance	2308 (39.7%)	1574 (23.6%)	
Self-pay	275 (4.74%)	254 (3.81%)	
No charge	28 (0.48%)	19 (0.30%)	
Other	280 (4.81%)	181 (2.72%)	
Medical comorbidities			
Hypertension	3238 (55.5%)	3801 (56.9%)	0.51
Diabetes mellitus	1224 (20.9%)	1993 (29.8%)	<0.0001
Congestive heart failure	533 (9.14%)	823 (12.3%)	0.013
Atrial fibrillation	741 (12.7%)	969 (14.5%)	0.19
Coagulopathy	727 (12.5%)	1009 (15.1%)	0.06
Chronic lung disease	1049 (18.0%)	1497 (22.4%)	0.01
Dyslipidemia	1140 (19.5%)	1570 (23.5%)	0.02
Chronic renal failure	411 (7.05%)	481 (7.20%)	0.89
Smoking	683 (11.7%)	342. (5.13%)	<0.0001
Alcohol abuse	205 (3.52%)	24 (0.37%)	<0.0001
Hospital size (%)			
Small	363 (6.26%)	461 (6.92%)	0.51
Medium	1295 (23.3)	1365 (20.5%)	
Large	4142 (71.4%)	4838 (72.6%)	
Hospital location and teaching status (%)			
Rural	227 (3.91)	174 (2.62%)	0.01
Urban nonteaching	1829 (31.5%)	1799 (30.0%)	
Urban teaching	3744 (64.5)	4691 (70.4%)	
Hospital region (%)			
Northeast	1043 (17.9%)	1452 (21.7%)	0.001
Midwest	1191 (20.4%)	1389 (20.8%)	
South	2199 (37.7%)	2678 (40.1%)	
West	1401 (24.0%)	1165 (17.4%)	
APRDRG Severity			
None to minimal loss of function	322. (5.53%)	188 (2.82%)	<0.0001
Moderate loss of function	1187 (20.3%)	2042 (30.5%)	
Severe loss of function	4324 (74.1%)	4453 (66.6%)	

**Table 2 medicina-55-00375-t002:** In-hospital complications, treatments, and outcomes of mechanically ventilated patients with Guillain–Barré syndrome (GBS) and myasthenia gravis (MG). PLEX, plasma exchange; IVIG, intravenous immunoglobulin.

	Mechanically Ventilated GBS	Mechanically Ventilated MG	*p*-Value
In-hospital complications			
Sepsis	1013 (17.4%)	784 (11.7%)	<0.0001
Pneumonia	1630 (27.9%)	1084 (16.2%)	<0.0001
Urinary tract infection	1888 (32.4%)	1585 (23.7%)	<0.0001
Procedures			
Gastrostomy	1950 (33.4%)	797 (11.9%)	<0.0001
Treatment (IVIG alone PLEX alone or combination)	3259 (55.8%)	3886 (58.1%)	0.24
Transfusions	970 (16.6%)	875 (13.1%)	0.02
Hospital charges ($) (mean 95% CI)	298,378 (±253,167)	176,220 (±163,584)	<0.0001
Outcomes			
Length of hospital stay (days) (mean 95% CI)	25.6 (± 21.8)	15.7 (± 12.9)	0.001
Prolonged length of stay (≥7 days) 77 days)	5576 (95.6%)	5515 (82.5%)	<0.0001
Alive at discharge			
	5223	6096	
None to minimal disability	878 (16.8%)	3942 (64.6%)	<0.0001
Moderate to severe disability	4345 (83.2%)	2154 (35.3%)	
All patients			
In-hospital mortality	586 (10.0%)	564 (8.45%)	0.16

**Table 3 medicina-55-00375-t003:** Multivariate analysis of the outcomes.

**Model 1**				
**Analysis Comprising All Alive Patients**				
	Unadjusted		Adjusted for Age, Gender, and Potential Confounders *	
	OR (95% CI)	*p*-Value	OR (95% CI)	*p*-Value
None-to-minimal disability	Reference			
Moderate-to-severe disability	9.05 (7.22–11.3)	<0.0001	15.6 (10.9–22.2)	<0.0001
**Model 2**				
**Analysis Comprising All Patients**				
	Unadjusted		Adjusted for Age, Gender, and Potential Confounders *	
Outcomes	OR (95% CI)	*p*-value	OR (95% CI)	*p*-value
Prolonged length of stay (≥ 7 days)	4.58 (3.33–6.31)	<0.0001	3.48 (2.22–5.48)	<0.0001

* Model adjusted for age, gender, significantly different medical comorbidities, in-hospital complications, and IVIG ± PLEX. OR = 1, no differences; OR > 1, the condition is more likely to occur in the GBS group; OR < 1, the condition is less likely to occur in the GBS group.

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
