# Peer review of "Comparative Outcomes of Respiratory Failure Associated with Common Neuromuscular Emergencies: Myasthenia Gravis versus Guillain–Barré Syndrome"

_medicina, 2019, doi:10.3390/medicina55070375_

Round 1
Reviewer 1 Report
Line 156." Both treatments are equally effective and the combination has neither ................" Authors should explain how did they come to this conclusion from database analysis. As treatment effects cannot be confirmed from this database.
Conclusion. I suggest authors specify that the conclusion are drawn from " NIS database" although they have stated clearly in the limitation of the study section.
Author Response
Respected reviewer,
Line 156
GBS in Adults:
AAN Therapeutics & Tech Subcommittee. Neurology. 78; 1009; 2012.
• Based on 2 Class I studies, IVIG is as efficacious as plasmapheresis for treating GBS in adults.
• Based on one adequately powered Class I study, the combination of plasmapheresis and IVIG is
probably not better than either treatment alone.
Comparison of IVIG & Plex in MG
Barth, et al Neurology 2011;76
Improved: 69% IVIG and 65% PE
Conclusion: IVIG & PE both effective Rx
As it is well established in prior studies that both treatments are equally effective, in our study we mainly concentrated on the outcomes in patients especially with respiratory failure associated with Myasthenia Gravis versus Guillain-Barré Syndrome. And also it would be difficult to segregate the patients who exclusively received IV Ig or Plasmapheresis or both from NIS data set.
Conclusion: I have included that the conclusions are drawn from NIS data
Reviewer 2 Report
The authors presented the topic that is of interest both to clinitians and scientists. The findings will further improve clinical practice and clinical decision making strategies.
Well done.
The authors conducted the retrospective analysis: Comparative outcomes of respiratory failure associated with common neuromuscular emergencies: Myasthenia Gravis versus Guillain-Barré Syndrome, and compared mechanically ventilated patients with MG and GBS at the national level in terms of in-hospital complications, length of stay, disability at discharge and in-hospital mortality.
The Introduction section is informative and proposed aim is sound.
The Methods section was adequately divided, the study parameters (variables) were adequately chosen and presented, with satisfactory exclusion criteria.
The Results section is understandable, presented in clear manner. Tables are informative. However, the first sentence in the Results section is probably typo error and I think that it was meant to be subheading, thus it should be corrected.
The Discussion section - authors stressed importance of mechanical ventilation both in MG and GBS groups of affected individuals. They further discussed their findings, stating that GBS patients have higher in hospital complications, LOS, higher costs and higher disability. They tried to explain such findings by the support of the premise that axonal lesions in some types of GBS are leading to greater functional impairments.
Study limitations were completely and adequately presented.
The main strength of this study aside the obtained results that will be of great importance for further clinical practice and further research, is large study population from adequate database pointing to current clinical management of mechanically ventilated MG and GBS patients.
Therefore, given the above all, it is my pleasure to suggest Acceptance of this manuscript for publication.
Author Response
Respected reviewer,
I have corrected the typo error in the result section.
Appreciate your time and effort to review our paper.
Thanks
Reviewer 3 Report
Authors have compared the respiratory failure in patients of MG and GBS. Since both the diseases have many similar symptoms, it is important to understand the differences in order to be prepared for associated complications. The review is written in a clear and concise language which would be easier for the readers to understand.
Authors can include following additional points and address following questions:
What is the need for this study please add in introduction and discussion. Write in general and in brief about pathophysiology of both diseases and criteria for intubation. It would be important to include the pulmonary function testing that could help determine the complications. What could be the essential predictors of respiratory failure in individual disease and would there be any differences?
Author Response
Dear reviewer,
I appreciate your time and effort to review this paper.
The objective of this study is to compare the outcomes especially in those patients on mechanical ventilation from these two common neuromuscular emergencies. This has been included in the introduction section.
As this is an observational study, with due respect the authors felt that inclusion of the pathophysiology especially with different variants for the diseases may make the paper very lengthy and ultimately tone down the message of this paper.
Due to the limitations of the NIS data is not possible to extract the information regarding pulmonary function testing and predictors of respiratory failure. But the point well taken.
Thanks